# Motivation-Aware Session Planning over Heterogeneous Social Platforms

Submission Id: 919

## ABSTRACT

With the explosive growth of online service platforms, an increasing number of people and enterprises are undertaking personal and professional tasks online. In real applications such as trip planning and online marketing, planning sessions for a sequence of activities or services will enable social users to receive the optimal services, improving their experience and reducing the cost of their activities. These online platforms are heterogeneous, including different types of services with different attributes. However, the problem of session planning over heterogeneous platforms has not been studied so far. In this paper, we propose a Motivation-Aware Session Planning (MASP) framework for session planning over heterogeneous social platforms. Specifically, we first propose a novel HeterBERT model to handle the heterogeneity of items at both type and attribute levels. Then, we propose to predict user preference using the motivations behind user activities. Finally, we propose an algorithm together with its optimisations for efficient session generation. The extensive tests prove the high effectiveness and efficiency of MASP.

## CCS CONCEPTS

• Information systems → Decision support systems; **Personalization**.

## KEYWORDS

Session planning, Heterogeneous social platform

**ACM Reference Format:**

Anonymous Author(s). 2018. Motivation-Aware Session Planning over Heterogeneous Social Platforms. In *Proceedings of Make sure to enter the correct conference title from your rights confirmation emai (WWW '2025)*. ACM, New York, NY, USA, 12 pages. https://doi.org/XXXXXXX.XXXXXXX

## 1 INTRODUCTION

The popularity of online service platforms has provided a vital channel for people and enterprises to undertake personal and professional activities online. Recent statistics show there are now 2.8 million active Australians on TripAdvisor and 1.5 million users on Yelp [1]. Users get access to these online service platforms for various purposes such as trip planning and online purchase. These

---

[1]https://www.socialmedianews.com.au

have given rise to a demand for assisting users in planning sessions of activities they wish to engage in. Popular platforms like Meituan and Google Maps provide services to numerous travellers for information on points of interest, as they offer item details and recommendations. According to EnterpriseAppsToday [2], Google Maps locates hundreds of millions of places and businesses. More than a billion people use Google Maps every month to search for destinations and check the best routes. However, these platforms only recommend a list of items based on item type or keywords. In practice, users could set up a set of activities and require a detailed travel plan. Take travel planning as an example as shown in Figure 1c. The user gives a set of interested activities and the system provides a plan that contains exact items and corresponding time. Thus, designing advanced session planning solutions becomes a new research problem and is promising for improving the service quality of these platforms, and improving their user experience.

Session planning has contexts and objectives that are different from those for Session-Based Recommender Systems (SBRS) and Personalized Route Planning (PRP). Traditionally, SBRS [10, 14, 16, 29, 41] predict the next item or item session based on historical sessions. PRP methods [5, 21, 31] typically generate user-specific routes in response to users' queries, considering user preferences and other factors like checkpoints or distance constraints. However, in session planning, users provide multiple activity categories, resembling a set of item categories, such that the system predicts the optimal session plans for their future actions based on historical item sessions. A well-generated session should align with the user preferences and certain related constraints. Figure 1 shows examples of SBRS, PRP, and session planning. Suppose a user $u_i$ named Jack recently visits 7-Eleven from home, and his profile keeps his historical activity sessions as shown in Figure 1 (a). SBRS would suggest McDonald's and ANZ Bank since his profile keeps a historical session of activities, *Home, 7-Eleven, McDonald's, ANZ Bank*. As shown in Figure 1 (b), Jack wants to travel from home to *The Flyfisher*, with two checkpoints, *IGA Hawthorn* and *QV Melbourne*. PRP would provide several routes that pass through all the POIs, based on his historical routes, considering different modes of transport, travel time, and transportation costs. While Jack would like to go to QV Melbourne from home and wants to conduct four activities: shopping, refuelling, having lunch, and finding a parking lot as shown in Figure 1 (c), he needs the system to plan his activities. The session planning could generate a series of specific POIs, *7-Eleven, Wilson Parking, Qv Melbourne, Grill'd QV*, for his planned activities.

This paper proposes a *Session Planning* over *Heterogeneous social Platform* (SPHP) problem, where users provide sets of activity categories and request ordered item sequences. The platforms could provide heterogeneous services (items) to users, and users can request multiple services. However, a big challenge is that we

---

[2]https://www.enterpriseappstoday.com/

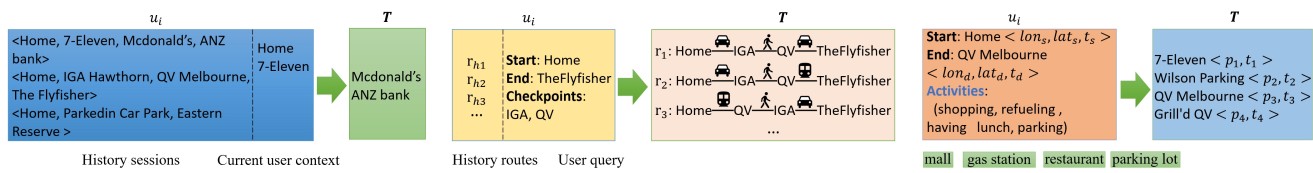

**Figure 1: Session recommendation vs. personalized route planning vs. session planning**

cannot predict an optimal session if only the contexts (source and destination) and objectives (service/activity categories) are provided. This is because a user may have dynamic preference for each activity category and the activity categories in a session are usually unrelated with each other. As shown in Figure 1c, the activity categories, *refueling* and *having lunch*, have no connections for current user session prediction. Thus, a system generated plan may not reflect user's dynamic preferences on activities and overall interests with respect to the time and distance constraints of the session. **How can we predict the optimal activity sessions without knowing the dynamic preferences of users?** Another challenge is that the item variety on platforms causes data heterogeneity at both attribute and type levels. Unlike SBRS where a session includes highly related items or user behaviours of the same type, the items within a session can vary significantly from each other in SPHP. As shown in Figure 1c, the gas station 7-Eleven and the restaurant Grill'd QV are two different types of establishments, each with distinct services. Existing solutions for heterogeneity problem take item attributes as auxiliary information [28] or describe each item as an attribute set [18], which generates item embeddings with extremely high dimensionality and ignores the correlation between attributes and types. Table 1 shows three items with their attributes from Yelp. Clinic 1 and Restaurant 1 belong to different categories and have different attributes. Restaurant 1 and Restaurant 2 belong to the same category, but still have different attributes such as Alcohol and GoodForKids. **How can we model the heterogeneous items with variable and large number of attributes and types?** SPHP is a problem of global optimisation over all the services on heterogeneous platforms. Since the volume of services is huge, we have a further challenge on real-time response of the system. **How can we quickly identify the activity sessions over a huge number of heterogeneous online services?**

| Items | Attributes |
|---|---|
| Clinic 1 | Accepts Insurance; By Appointment Only; Business Accepts Bitcoin. |
| Restaurants 1 | Good For Meal; WiFi; Good For Kids; Has TV; Restaurants Reservations; Business Parking. |
| Restaurants 2 | Restaurants Attire; Business Accepts CreditCards; Alcohol; Good For Kids; Restaurants Reservations; Business Parking; Bike Parking; Restaurants Delivery. |

**Table 1: Variety of items.**

This paper proposes Motivation-Aware Session Planning (MASP) framework that fully exploits the driving force behind activities to predict the user preferences for SPHP. We first propose a novel HeterBERT model to capture the attribute-level and type-level heterogeneity. Then, we design a motivation-aware solution to generate motivation-aware item/user embeddings. Final plans are obtained by multi-constraints session generation.

- We propose a novel session planning over heterogeneous social platform (SPHP) problem, which generates optimal sessions for users requesting services.

- We propose a novel HeterBERT model to address the attribute-level heterogeneity. HeterBERT well handles the problem of uncertain attribute number of items and captures the type-attribute correlation in heterogeneous items.
- We design a motivation-aware prediction solution that fully exploits the motivations behind activities to capture the dynamic preferences of user for session planning.
- We propose a multi-constraints session generation algorithm together with optimisation strategies that enables effective and efficient multi-constraints session generation. The test results prove the performance of MASP.

## 2 RELATED WORK

This research is relevant to session-based recommendation, personalized route planning and heterogeneous social media processing.

SBRS learn users' preferences from the sessions associated and generated during the consumption process. Each session includes multiple user-item interactions occurring together over a period, typically lasting for up to several hours. Conventional SBRSs [8, 15] employ data mining or machine learning techniques to capture the dependencies embedded in sessions for recommendations. Latent representation-based SBRS [13, 20] construct a low-dimensional latent representation for each interaction within sessions using shallow models for recommendation. Recently, DNN-based SBRSs [3, 17, 27, 34, 39] has been popular due to their powerful capabilities to model the complex intra-session and inter-session dependencies. In [17], a session graph and its mirror graph are constructed, and the information propagation between them is conducted with an iterative dual refinement for representation learning. GRec [39] adopts CNN with sparse kernels for item and session embeddings. SEOL [27] enhances recommendation using session tokens, session segment embeddings, and temporal self-attention. KMVG [3] learns three item representations from the knowledge graph, contextual transitions in sessions, and local item-item relationships, which are merged as the final item representation. In SPHP, a session is an activity plan that includes items to interact with a user shortly. However, in SBRS, a session refers to a series of history or current user behaviours in a period. Thus, SPHP is a new research problem. The SBRS methods cannot be applied or extended for SPHP.

Route planning aims to generate the top $k$ probable routes that satisfy a query containing the origin, destination, a set of checkpoints, a maximum time cost, etc. Conventional methods [2, 5, 42] aim to find the optimal routes according to a specified objective, such as minimizing distance, time, or cost. Recent deep learning (DL)-based methods [1, 21, 23, 24, 31, 35] prevail in route planning since they can discover complex relationships among data. NASR+ [31] models the observable trajectory by attention-based RNNs and estimates the future cost using position-aware graph attention

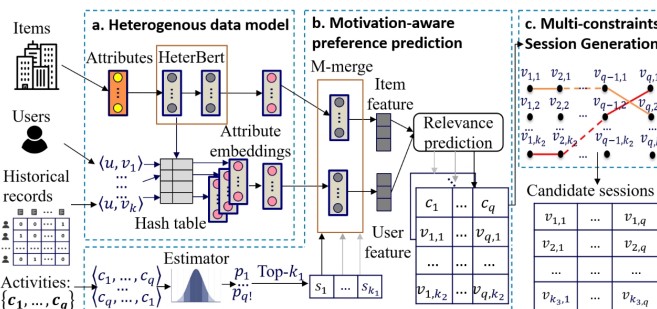

**Figure 2: Overview of MASP framework.**

networks. SpeakNav [1] exploits BERT to extract clues from user speeches for generating routes. MaMoRL [24] adopts a multi-agent multi-objective reinforcement learning framework. VIAL [35] enhances A* algorithm with a variational inference-based estimator to model the distribution of travel time between two nodes. MAC [23] learns the knowledge from geographical and semantic neighbours to be combined for predicting the next item. In SPHP, a user keeps implicit objectives and only gives a set of activities, while PRP methods require explicit objectives like checkpoints and transport modes. In addition, existing PRP [21, 22, 40] only handle heterogeneous data from item level and interaction level, while cannot handle attribute-level heterogeneity in SPHP.

Heterogeneous social media has been handled using cross-domain and transfer learning (TL)-based methods. Cross-domain methods [11, 12, 26, 36] address the data heterogeneity using the information from multiple domains or sources. For example, GCBAN [11] embeds items/users and their auxiliary information into two latent spaces for each data domain. Two types of latent features are concatenated and applied to a Gaussian-based probabilistic model for recommendation. EquiTensors [36] aligns heterogeneous datasets to a consistent spatio-temporal domain, and learns shared representations using convolutional denoising autoencoders. HetSANN [12] constructs a heterogeneous user-item graph. An attention mechanism learns the node embeddings for node classification. However, HetSANN requires uniform attributes, thus inapplicable to handling the attribute-level heterogeneity in SPHP. TL-based models [19, 32] exploit the knowledge gained from a previous task to generalize the model for other tasks. DDTCDR [19] transfers information between two types of items by dual learning. BALANCE [32] obtains knowledge from dynamic and heterogeneous workloads by transfer reinforcement learning. In SPHP, the number of attributes is variable and large, causing attribute-level heterogeneity. However, none of existing PRP can handle the attribute-level heterogeneity.

## 3 PROBLEM FORMULATION

This section defines the concepts of item, activity, and session, and formally formulates the SPHP problem.

DEFINITION 1. *In heterogeneous social platforms, an **item** refers to a real-world entity, such as a restaurant or store, providing a type of service at a specific location and time period. An **activity** refers to a particular type of user behaviour or action such as shopping, park visits, or medical appointments. Each activity is taken by users through the platform's service. A user may request multiple services from different items within a time period.*

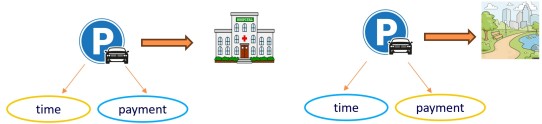

**Figure 3: Motivations behind activities.**

Each item has several content attributes that describe its properties, and two contextual attributes, geographic location and opening hours. An item can have up to 33 attributes. The heterogeneous items present special characteristics in contrast to general items.

- Attribute-level heterogeneity: The number of attributes in each heterogeneous item can be large and different items may have different numbers of attributes.
- Type-level heterogeneity: A social platform includes various item types such as "restaurant" and "hotel". Different types normally share few common attributes.
- Type-attribute correlation: Attributes are associated with types. Even if some attributes from different item types share common names (e.g. "price" in "hotel" and "restaurant"), they reflect different semantic meanings.

In practice, a user may take a series of activities within a time period. A number of items, each of which is associated with an activity, are arranged in order to form a session.

DEFINITION 2. *A **session** is a service plan for a user including a series of items to be interacted by the user. Formally, a session is a series $<v_1, \cdots, v_q>$, where $v_i$ is the $i^{th}$ item and $q$ is the session length.*

Given a user and a set of his/her planned activities, an ideal session should contain the items that provide these service activities, satisfying the user's preference, under the context constraints. The problem of *SPHP* is formally defined as follows:

DEFINITION 3. *Given a user $u$, a set of activities $\{c_1, \cdots, c_q\}$, and a session score function $f_{score}$, SPHP aims to detect a list of sessions with the highest probability scores, satisfying the constraints below:*

- *Distance constraint: a user can only visit the items that are within the radius of this user.*
- *Time constraint: each item has available time and a user can only visit the available items.*

We address the problem of effective and efficient SPHP, and propose a motivation-aware session planning framework (MASP) for SPHP. Here, ***motivation*** refers to the internal or external driving force, such as seeing doctor or sales promotion, that drive individuals to undertake specific actions. Motivation may stem from users' needs, desires, goals, or external stimuli, which guide user behaviours to fulfil their objectives. When planning activities, individuals typically align their behaviours with their underlying motivations to achieve their goals. Figure 2 shows MASP that mainly contains three parts: heterogeneous data model, motivation-aware preference prediction, and multi-constraints session generation.

## 4 MOTIVATION-AWARE SESSION PLANNING

Intuitively, a set of activities could imply users' motivations which lead to different strategies when selecting individual activities. Figure 3 shows two activity sets <parking, go to hospital> and <parking, go hiking>. When users are seeking medical care, they may prioritize hospitals with closer parking facilities. However,

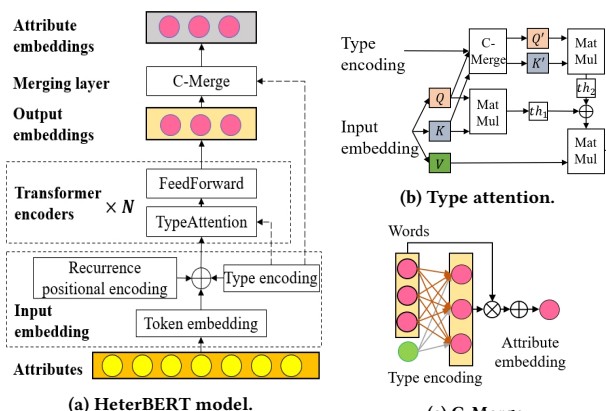

**(a) HeterBERT model.**

**(b) Type attention.**

**(c) C-Merge.**

**Figure 4: HeterBERT.**

when heading out for outdoor activities, they might value parking lots that are more affordable. This indicates that motivations behind activities could influence users' decisions.

## 4.1 Heterogeneous Data Model

We build a model to discover the heterogeneous item types with an uncertain number of correlated attributes.

*4.1.1 HeterBERT.* Each heterogeneous item on social platforms can be described as a set of attributes, and different items can have different numbers of attributes. To represent a heterogeneous item, one-hot encoding takes the item attributes as auxiliary information [28]. Other approaches represent each item as a set of attributes [18]. With these methods, the dimensionality of an item embedding equals the number of attributes. The total number of attributes could be huge, which leads to a large size of one-hot encoding and incurs the curse of dimensionality, further incurring high memory costs for data storage and high time costs for data updates introduced by new items or attributes. In addition, all existing methods ignore the correlation between attributes and types, which fails to capture the item attributes under different types. BERT-based model is suitable for text embedding learning, which addresses the uncertain attribute number problem by the BERT padding operations. By dividing attributes into words, BERT can learn word embedding and indirectly learn attribute embedding as item representation. However, turning each attribute sentence into separate words, the BERT-based model cannot capture the word correlation in the same attribute, and thus cannot address the attribute heterogeneity of items. In addition, BERT cannot handle type-level heterogeneity or capture the type-attribute correlation since it ignores the type information. For example, the word "insurance" carries completely different meanings when it comes to hotels and clinics. We need to build a model that is robust to the heterogeneity of items at attribute and type levels and captures the type-attribute correlation.

We propose a novel HeterBERT model for heterogeneous items, as shown in Figure 4. HeterBERT advances BERT [7] in fourfold: (1) HeterBERT adopts our new proposed recurrence positional encoding and type encoding to keep the correlation among inner-attribute words and capture the type information respectively; (2) a new type attention layer is designed to inject type into word embedding learning, with a double threshold mechanism to enhance the correlation among inner-attribute words. (3) a C-Merge layer is

| #cls | restaurant | take | out | #sep | restaurant | park | $\cdots$ |
|------|-----------|------|-----|------|-----------|------|----------|
| 0 | 1 | 2 | 3 | 0 | 1 | 2 | $\cdots$ |
| $E_0$ | $E_1$ | $E_2$ | $E_3$ | $E_0$ | $E_1$ | $E_2$ | $\cdots$ |

**Figure 5: Recurrence positional encoding.**

$$\text{Types} \longrightarrow \boxed{\text{Token embeddings } E} \xrightarrow{\{W_c\}} E_C \xrightarrow{W} \bar{E}_C$$

**Figure 6: Type encoding model.**

proposed to capture the type-attribute correlation; (4) considering the type-level heterogeneity, a contrastive learning task is designed to learn discriminative embeddings.

**Input Embedding.** Given an item, the input embedding layer divides the item attributes into words and generates an initial embedding for each word. To address the heterogeneity, we propose a recurrence positional encoding which considers the relative position of the words within each attribute while ignoring the relative position of attributes. Figure 5 shows an example of this recurrence positional encoding. When the special token *#sep* emerges, we reset the index of tokens. The positional embedding is generated by the sine and cosine functions as in [30]. By this encoding, the generated word embeddings can capture correlation among words in the same attribute, while allowing the attribute orders to be swapped for fitting the characteristics of the item attributes semantically.

To capture type information in embedding learning, intuitively, type vectors should be distinct from each other to make word embeddings discriminative from the type aspect. We propose a type encoding model as Figure 6 shows. Each type is first divided into words and the corresponding token embeddings $E$ are integrated by linear transformation $e = W_c E$ to generate intermediate type representations $E_C = [e_1, \cdots, e_C]$, where $W_c$ is the transformation matrix. Though each type has a parameter matrix $W_c$, the complexity of this type encoding model is limited, since the number of types is not large and the model architecture is shallow. Another linear transformation $\bar{E}_C = W E_C$ is applied on $E_C$ to generate the final type representation. We aim to maximise the difference among type representations and formulate the loss function as below:

$$Loss = 1/Var(\bar{E}_C), \tag{1}$$

where $Var(\bar{E}_C)$ is the variance of matrix $\bar{E}_C$. This loss function is supported by Theorem 1 proved in Appendix A.1:

THEOREM 1. *Given a set of type representations with mean value $\mu$, the distinct difference among these representations is achieved when $Var(\bar{E}_C)$ is maximised.*

The loss function has no regularization term, as we train this model over all types and directly use $\bar{E}_C$ as the type representations, which avoids the overfitting issue. Given an item, we generate the initial embeddings by summing the corresponding token embedding [30], recurrence positional encoding, and type encoding.

**Type Attention.** Given type representations and initial word embeddings, we feed them into the transformer layer. As shown in Figure 4b (we omit softmax, scaling layers, residual connection, and normalization layers for convenience), input embeddings are projected into word-level query, key, and value matrices $Q, K$, and $V$ by learned linear transformation matrices. Unlike the vanilla attention

layer that only feeds $Q$ and $K$ into the matrix multiplication layer to compute the word-level attention weights, we design C-Merge to capture the correlation among inner-attribute words, as Figure 4c shows. We merge word- and attribute-level attention weights, capturing the correlation among inter-attribute words. Specifically, given a word-level query matrix $Q$, a key matrix $K$, and type representation $\bar{E}_C$, we first divide $Q$ and $K$ into subsets and feed them into the C-Merge layer, each of which contains word embeddings to the same attribute. Given a subset, the C-Merge layer computes inner-attribute word weights by $W_{att} = HE$, where $E$ is the concatenation of word embeddings and the item type representation, $H$ is the transformation matrix. Then, we can get the attribute embedding by merging word embeddings $e_{att} = W_{att}\bar{E}^T$, where $\bar{E}$ contains word embeddings. Especially, the C-Merge layers of each type attention layer share parameters and type representation to reduce the computation time and maintain type information in the whole embedding learning. The attribute-level query and key matrices $Q'$ and $K'$ are generated by combining all attribute embeddings from $Q$ and $K$ respectively. Given attribute-level $Q'$ and $K'$, we compute attribute-level attention weights by the dot product of these matrices. We select the word- and attribute-level weights larger than their thresholds to enhance the correlation among words within the same attribute. Then the selected weights are merged by weighted sum to generate the new word-level weights:

$$W^* = w_T W_{T1} + (1 - w_T)W_{T2}, \qquad (2)$$

where $W_{T1}$ and $W_{T2}$ are filtered weights and $w_T$ is a trade-off parameter. The attention layer feeds the value matrix $V$ and the new word-level weights into the matrix multiplication layer to derive the updated word embeddings. The output word embeddings are fed into a position-wise fully connected feed-forward network with a residual connection normalization [30]. The feed-forward network produces updated word embeddings for the next attention layer.

Given the output word embeddings of the last attention layer, we feed them into a C-Merge layer in HeterBERT (Figure 4a) to get attribute embeddings, injecting the type into attribute embeddings and capturing the type-attribute correlation.

**Pre-training HeterBERT.** To save the training cost, we pre-train HeterBERT by two tasks: Masked Language Model (MLM) and Contrastive Learning (CL). For MLM, we randomly select 15% of the words from all attributes, adopt a masking procedure, and predict the masked tokens, as in [7]. Specifically, we use the BERT masking which replaces 80% of the selected words with *mask* token, replaces 10% of those with a random word, and keeps the rest 10% of those unchanged. For example, given an input "Restaurant Take Out" where "Out" is the selected word, there are three masked inputs: "Restaurant Take [mask]", "Restaurant Take WEB", and "Restaurant Take Out". By applying softmax to the output embeddings at the positions of the masked tokens, the prediction results can be obtained. As in [7], we use the categorical cross-entropy loss $L_{MLM}$.

For the second learning task, the ideal attribute embeddings should be distinct when these attributes belong to different types, due to the type-level heterogeneity. These embeddings should keep the diverse information of attributes belonging to the same type, as these attributes may describe different items from different aspects. Unlike BERT that outputs two sentences from two segments of input tokens, HeterBERT outputs $l$ sentences to attribute embeddings,

where $l$ is the number of attributes in an item. Thus, Next Sentence Prediction (NSP) used by BERT, is improper for HeterBERT training. Other supervised learning tasks like type classification are also unsuitable since they could make attributes in an item excessively similar in feature space. We propose a CL task for training Heter-BERT by using "positive" and "negative" data. Specifically, we first randomly select $n_p$ pairs of attribute embeddings from the same items as "positive" pairs. Then, we derive $n_n$ "negative" pairs by selecting two attribute embeddings from two items. We assume the distance of "positive" pairs is smaller than "negative" pairs. We formulate the loss function as follows:

$$L_{CL} = -ln(\sum_{p_i^+}^{n_p} \phi(e_j^+, e_k^+) - \sum_{p_i^-}^{n_n} \phi(e_j^+, e_k^-)), \qquad (3)$$

where $\phi(\cdot)$ is cosine similarity, $p_i^+$ is the $i$-th "positive" pair $(e_j^+, e_k^+)$, and $p_i^-$ is the $i$-th "negative" pair $(e_j^+, e_k^-)$. With HeterBERT contrastive learning, we learn the attribute embeddings that are distinct at the type level, while keeping the diverse information of attributes.

*4.1.2 User Profile Construction.* Each social user contains many types of data, like friendship and interaction records. We construct the profile based on her/his interacted items and friends on social platforms. Given a user $u$, s/he should be interested in some attributes of the interacted items, and s/he should share some common interests with her/his friends. Thus, we first construct a historical attribute set $A_u$ by collecting attributes from interacted items and form a neighbour attribute set $A_n$ by combining historical attribute sets $A_u$ of the user's friends. Then, we combine these sets as $\bar{A}_u$ to reflect which attributes the user is interested in. The attribute embeddings of $\bar{A}_u$ form the user profile as $E_u$. Formally, a user profile is described as 6-tuple $U = <uid, I, N, A_u, A_n, E_u>$, where $uid$ is the user id; $I$ is the history set, containing $vid$ of interacted items; $N$ is the neighbour set, containing $uid$ of $u$'s friends; $A_u$ is the attribute set of interacted items; $A_n$ is the attribute set, combining $A_u$ sets of $u$'s friends; $E_u$ is the attribute embedding set, combining the corresponding attribute embeddings of $A_u \cup A_n$.

## 4.2 Motivation-aware Preference Prediction

We build a model that considers motivation and dynamically merges attribute embeddings of users and items.

*4.2.1 Activity Category Arrangement Algorithm.* Given a activity set, we need to generate all the possible arrangements. However, in practice, some arrangements are not reasonable. For example, given an activity set {$Parking, Restaurant, Shop$}, an arrangement $< Restaurant, Parking, Shop >$ is unreasonable since people often park before doing other activities. Thus, we propose an Activity Category Arrangement (ACA) algorithm to select candidate category arrangements. The algorithm is detailed in Appendix A.2.

*4.2.2 M-Merge User Preference Prediction Model.* To predict user preference, we need to represent users and items from attribute embeddings into uniform features. Existing methods [28] represent an item by merging its attributes with concatenation or weighted summing, which incurs redundancy from similar attributes and weaken the influence of key attributes. As Figure 3 shows, under different motivations, user preference could vary for the same type of items. We propose a motivation-based merge (M-Merge) layer,

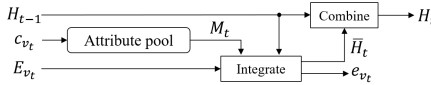

**Figure 7: M-Merge model.**

as shown in Figure 7, to dynamically generate user and item representations under the motivation behind an arrangement. Then, the user-item relevance is computed for candidate item list generation.

**Motivation-aware Item Representation.** Given an item, M-Merge generates an item feature by merging its attribute embeddings under the motivation behind a category arrangement. M-Merge needs to capture the influence of previous activities in item representation, due to the contextual correlation from activity series. Given a set of historical attribute embeddings $H_{t-1}$, the current category $c_{v_t}$, and attribute embeddings of the current item $E_{v_t}$, M-Merge integrates attribute embeddings based on the current motivation and previous items. Specifically, given a $c_{v_t}$, we sample $N_M$ attribute embeddings from the corresponding attribute pool, as the motivation set $M_t \in R^{N_M \times d}$, based on their occurrence frequency ($N_M$ is set to 20 empirically[3]). We construct an attribute pool for each type by gathering attribute embeddings from items to that type. Given the historical attribute embedding set $H_{t-1}$, the current motivation set $M_t$, and the current item attribute embedding set $E_{v_t}$, the M-Merge layer first measures how much an attribute meets another attribute from the current motivation/previous interaction by the dot product of two attribute embeddings. Extending to the whole input $E_{v_t}$, we formulate two similarity matrices $E_{v_t}M_t^T$ and $E_{v_t}H_{t-1}^T$. Then, we apply *RowSum* on these matrices, followed by *SoftMax* and *Transposition*, to generate weights $W_M \in R^{1 \times N_E}$ and $W_H \in R^{1 \times N_E}$. The integration is formulated by:

$$e_{v_t} = Softmax(\omega_v W_M + (1 - \omega_v)W_H)E_{v_t}, \quad (4)$$

where $\omega_v$ is a trade-off parameter. We regard the integrated weights as corresponding probabilities for item attribute embeddings. Then, we repeatedly sample an attribute from $E_{v_t}$ at the probability distribution and add it to $\overline{H}_t$ until $|\overline{H}_t| > p_{rop}|E_{v_t}|$, where $p_{rop}$ is a parameter which controls the number of sampling. $H_t \in R^{N_H \times d}$ is generated by combining the previous attributes $H_{t-1} \in R^{m_1 \times d}$ and current attributes $\overline{H}_t \in R^{m_2 \times d}$, where $N_H = m_1 + m_2$.

**Motivation-aware User Representation.** Similar to item representation, given a user, the model needs to capture the influence of previous activities in user representation and the current motivation. We use another M-Merge model to dynamically generate the user feature. Given a type $c_{v_t}$, we first randomly select $N_M$ attribute embeddings from the corresponding pool as the motivation set $M_t \in R^{N_M \times d}$. Given a set of user attribute embeddings for previous activities $G_{t-1}$, the current motivation $M_t$, and a user profile $E_u$, we then formulate two similarity matrices $E_u M_t^T$ and $E_u G_{t-1}^T$ and generate historical/current weights $W_M \in R^{1 \times N_E}$ and $W_G \in R^{1 \times N_E}$ by *RowSum*, *SoftMax*, and *Transposition* operations. The user feature is formulated by:

$$e_{u_t} = Softmax(\omega_u W_M + (1 - \omega_u)W_G)E_u, \quad (5)$$

where $\omega_u$ is a trade-off parameter. Like constructing current historical item attributes, we first sample attributes at the corresponding

[3]https://www.maitaowang.com/article/48685

probabilities to form $\overline{G}_t \in R^{n_1 \times d}$, then combine it with previous historical user attributes $G_{t-1} \in R^{n_2 \times d}$ to construct current historical user attributes $G_t \in R^{N_G \times d}$, where $N_G = n_1 + n_2$. The item and user representations are fed into the relevance prediction layer. Given a user feature $e_{u_t}$ and an item feature $e_{v_t}$, we define a user-item relevance $r_{u_t,v_t} = e_{v_t}e_{u_t}^T$. A larger $r$ means a user is more likely attracted by an item. Following [27, 41], we utilise AdamW as the optimiser and cross entropy as the loss function.

**Candidate Item List Generation.** Given a user $u$ and her candidate category arrangement $< c_1, \cdots, c_q >$, we compute user-item relevance scores by $q$ steps. When $t = 1$, we first feed the user, items belonging to $c_1$ and current type $c_1$ into two M-Merge models. Especially, the historical attributes for an item and a user $H_0$ and $G_0$ are initialized as empty sets. After we achieve user and item features, we select the top-$k_2$ items with high relevance scores into the candidate list $R_{c_1}$ for the first activity. In addition, for each selected item, we obtain a pair of current historical attributes set $\overline{H}_1$ and $\overline{G}_1$. Accordingly, we derive $H_1$ and $G_1$ by combining the corresponding sets as the input at step $t = 2$. Extending the process to step $t$, we generate a candidate item list $R_t$. For the given arrangement, we generate a set of item lists $R = < R_1, \cdots, R_q >$. The detailed algorithm ACA is shown in Appendix A.2. In practice, new items and session records could appear over platform. We dynamically and incrementally maintain the models of MASP to reflect the most recent social updates. We adopt fine-tune training strategy [33] to jointly update the HeterBERT and M-merge on the new data.

### 4.3 Multi-constraints Session Generation

We propose a naive multi-constraint session generation (MCSG), and optimizations for fast candidate item generation and MCSG.

*4.3.1 Naive MCSG.* With HeterBERT and motivation-aware preference prediction, we achieve candidate item lists for different categories under given arrangements. As Figure 2 (c) shows, for each candidate arrangement, we first regard its item lists as a $q$-partite graph to $q$ independent sets. The task of generating a session can be converted into finding a path through each set sequentially. Then, our goal is to generate sessions that align with users' interests, meet their requirements for activity arrangement, and avoid breaching constraints. We evaluate each path by the user preference, correlation of adjacent items, and constraints. Given a user $u$ and a path $p_a = < v_1, \cdots, v_q >$, the user preference is described by the relevance score $r_{u,v_i}$, the correlation is computed by $e_{v_i}^T e_{v_j}$, and constraints filters improper paths. The session score is computed as follows:

$$f_{score}(p_a) = \sum_{i=1}^{q} (r_{u,v_i}) + \sum_{j=1}^{q-1} (e_{v_j}^T e_{v_{j+1}}). \quad (6)$$

The MCSG includes path generation and evaluation. For each item list $R^{(i)}$, we generate possible paths based on its distance and time constraints. We compute the spherical distance of adjacent items, check the distance constraint ($rad = 1$ km) and the available time of these items, as in [4, 25]. We evaluate each path by Eq. 6 and select top-$K$ paths for the $i^{th}$ candidate arrangement.

*4.3.2 Optimisation.* We propose attribute-based candidate items generation to accelerate the item list generation and a greedy algorithm to accelerate the session generation.

**Optimising Item List Generation.** Given a user and a candidate arrangement, we compute the relevance score between the user and all the items. A naive way to reduce the time cost is to divide items by their types and only compute those conforming to the arrangement. Given $q$ types, suppose that each type includes $m$ items, and there are $k_1$ candidate arrangements for these items, the time cost of relevance score computation is $O(m * q * k_1)$. When the number of items is large, the time cost is high. To reduce the computation time, index-based methods are unsuitable, since users and items are dynamically represented under motivation in MASP. We need to design an algorithm to quickly generate candidate item lists. We propose Theorem 2 to identify the items with higher scores.

THEOREM 2. *Given a user $u$ and her motivation $M_t$ or historical interests $H_{t-1}$, items with attribute embeddings similar to embeddings in $M_t$ or $H_{t-1}$ receive higher relevance scores than items that do not possess such similarity.*

We prove Theorem 2 in Appendix A.3. With Theorem 2, we propose an item list generation (ILG) algorithm that selects candidate items efficiently as detailed in Appendix A.4.

**Optimising MCSG.** Given an item list set $R^{(i)}$, it contains $q$ item lists. Each list contains $k_2$ items, where $k_2$ is 1.5 times of the returned sessions to avoid falsely filtering the sessions to be returned. Due to the type-level heterogeneity, items belonging to different types could significantly vary from each other, leading to a small correlation between them ($e_{v_j}^T e_{v_{j+1}} \ll r_{u,v_j} + r_{u,v_{j+1}}$). The approximation of the session score is formulated as: $\bar{f}_{score}(p_a) \approx \sum_{i=1}^{q}(r_{u,v_i})$. By only considering the weight of nodes, the session generation is converted into a group Knapsack problem, which is NP-Hard. Thus, we propose a Greedy-MCSG as detailed in Appendix A.5 to address the problem by selecting $N_{approx}$ paths using the greedy algorithm.

## 5 EXPERIMENTAL EVALUATION

Due to the page limit, we present the additional experiments in Appendix A.7-9.

### 5.1 Experiment setup

We conduct experiments with three real datasets, Yelp [4], Yelp-L and Douban [5]. The Yelp dataset consists of consumer and business activity based on hundreds of millions of reviews and photos, and millions of daily consumer interactions in the Los Angeles area. Each business possesses rich attribute information. Yelp-L is a large-scale version of Yelp, featuring more users, items, and interactions in more regions. The Douban dataset contains user and event information. Each event is associated with time and location.

Following previous works [37, 38], we sort the interaction by time and take 80% as training data to train models and learn parameters. 10% is used as a validation set to tune hyperparameters. The rest of the data is regarded as the test data to evaluate our model. We compare our MASP with its variant MASP-OPT and two STOA approaches, GNNAutoScale [9] and TiCoSeRec [6]. GNNAutoScale scales message-passing graph neural networks to large-scale graphs for recommendation. TiCoSeRec explores the impact of time intervals on sequential recommendations. Classical methods discussed

---

[4]https://www.yelp.com/dataset
[5]https://www.douban.com/location

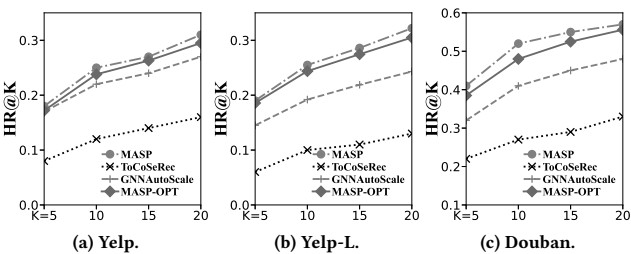

**(a) Yelp.**     **(b) Yelp-L.**     **(c) Douban.**

**Figure 8: Effectiveness comparison.**

**Table 2: Effect of incremental model update.**

| Updated data size | 20% | 40% | 60% | 80% | 100% |
|---|---|---|---|---|---|
| HR@20 for Yelp | 0.31 | 0.313 | 0.317 | 0.318 | 0.320 |
| HR@20 for Yelp-L | 0.322 | 0.325 | 0.328 | 0.331 | 0.333 |
| HR@20 for Douban | 0.572 | 0.573 | 0.577 | 0.578 | 0.579 |

in related work define sessions in a different way [17, 34, 39] or request additional information like checkpoints [1, 24, 35], which are unsuitable for SPHP. Thus, we do not include them in evaluation. The effectiveness is evaluated using Hit Ratio (HR@K) and Spearman's rank correlation, while efficiency is measured by response time. More details on experiment setup are in Appendix A.6.

### 5.2 Effectiveness Evaluation

*5.2.1 Effectiveness Comparison.* We compare four approaches, including two alternatives of our proposed models, MASP, MASP-OPT, and two SOTA techniques, GNNAutoScale and ToCoSeRec, in terms of $HR@K$ and report Spearman's rank correlation $\rho$ of MASP over three datasets. As Figure 8 shows, MASP achieves the best performance, followed by MASP-OPT. This is because MASP exploits the inter-item correlation and item-type correlation, thus the heterogeneous items with different types can be better distinguished in embedding space. Meanwhile, the M-Merge layers well capture the user motivations behind the activities, which further mines the correlations between heterogeneous items. With ILG and MCSG optimisations, the effectiveness of MASP-OPT drops slightly, since a small number of items are incorrectly filtered out. However, considering the tremendous improvements in efficiency, MASP-OPT is a good option for SPHP. Compared with ToCoSeRec, GNNAutoScale achieves better performance, due to the capability of learning item representation. ToCoSeRec performs worst since it follows the assumption that items within the same session are similar, which does not apply to the SPHP problem.

Furthermore, we compute Spearman's rank correlation for top-3 candidate category arrangements. The corresponding average correlation values are 0.940, 0.755, 0.597 for Yelp, 0.912, 0.7650, 0610 for Yelp-L, and 0.908, 0.770, 0.452 for Douban, which indicates our ACA is effective in selecting suitable candidate category arrangements.

*5.2.2 Effect of Model Update.* Following [43], we split the datasets into training, validation, new data, and test sets as 80%, 5%, 5%, and 10%. For the new data, we evaluate the performance of MASP across five batches of new data, which contains 20%, 40%, 60%, 80%, and 100% of the data. We incrementally update the models of MASP and use updated ones to predict the test data. Table. 2 shows the effect of the model update in term of HR@20. Clearly, with data updates, the effectiveness of MASP slightly increases for all datasets.

**Table 3: Optimisation comparison.**

| Time (s) | MASP | MASP-ILG | MASP-MCSG | MASP-OPT |
|---|---|---|---|---|
| Yelp | 21.2 | 4.5 | 19.5 | **3.9** |
| Yelp-L | 141.8 | 45.2 | 120.5 | **33.4** |
| Douban | 37.5 | 9.2 | 35.2 | **5.8** |

**Table 4: Efficiency comparison.**

| Time (s) | MASP | GNNAutoScale | ToCoSeRec | MASP-OPT |
|---|---|---|---|---|
| Yelp | 21.2 | 18.5 | 23.3 | **3.9** |
| Yelp-L | 141.8 | 122.1 | 158.4 | **33.4** |
| Douban | 37.5 | 32.2 | 45.7 | **5.8** |

## 5.3 Efficiency Evaluation

*5.3.1 Effect of Optimisation.* We first test the effect of the proposed optimisation and report the response time in Table 3. Clearly, the combination of two optimisations achieves the best efficiency, followed by ILG optimisation. This is because ILG could quickly filter out the items that do not share attributes with a target user. On the other hand, MCSG optimisation improves efficiency slightly. This is because Greedy-MCSG balances the information loss and efficiency, leading to a slight improvement in efficiency. MASP incurs the highest time cost, which proves the importance of optimisation.

*5.3.2 Efficiency Comparison.* We compare our MASP and MASP-OPT with SOTA methods, GNNAutoScale and ToCoSeRec, in terms of overall time cost. The the experimental results over three datasets are reported in Table 4. Clearly, our MASP-OPT is much faster than MASP, GNNAutoScale and ToCoSeRec. This is because the proposed ILG algorithm significantly reduces the candidate items for computing the relevance scores and Greedy-MCSG optimisation further accelerates the session generation. GNNAutoScale is slightly faster than MASP and ToCoSeRec because it generates sessions based only on item types, whereas MASP and ToCoSeRec consider both item types and correlations between items.

*5.3.3 Time Cost of Model Update.* We take the model update in two ways: fully re-training the learned model over new and old data and fine-tuning the trained model over new data. We split datasets into training, validation, new data, and test sets, as in Sec 5.2.2. We use 100% of new data to fine-tune the models of MASP for Yelp, Yelp-L, and Douban. For fine-tuning-based method, the time costs for the model update are 12.2 minutes, 50.7 minutes, and 31.4 minutes, respectively. For the re-training, it requires 16 hours, 65 hours, and 39 hours. Compared to re-training, the incremental strategy improves model update efficiency by up to about 75 times. Thus the time cost of model updates in MASP is well controlled.

## 5.4 Ablation Study

We conduct the ablation study to evaluate the impact of three main components of our MASP model: input embedding layer, C-Merge, double threshold mechanism, and M-Merge. Figures 9 -10 depict our ablation analysis's experimental results over three datasets.

*5.4.1 Input Embedding Layer.* We compare the HR@K results of MASP for two settings: the full version of the proposed input embedding (Full) and the naive input embedding of the original BERT (Naive embedding). Figure 9 displays the effectiveness comparison

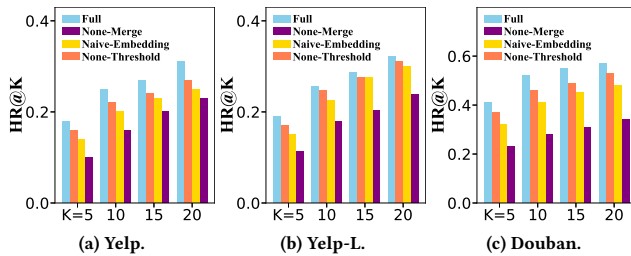

**Figure 9: Ablation study on input embedding, C-Merge, thresholds.**

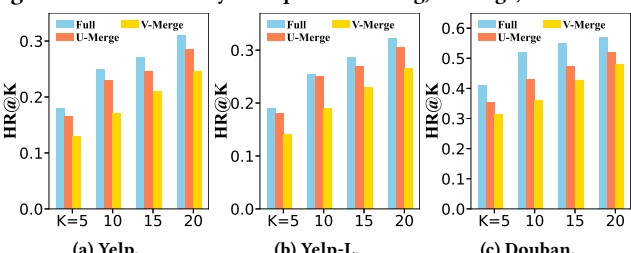

**Figure 10: Ablation study on M-Merge.**

results. With the proposed input embedding, MASP outperforms the model with naive input embedding across all K values, which indicates that our input embedding can handle the heterogeneity at attribute and type levels, thus enhancing the effectiveness of MASP.

*5.4.2 C-Merge Layer.* We compare the HR@K results of MASP obtained under two settings: full version (Full) and None C-Merge layers (None-Merge). As shown in Figure 9, the model's performance significantly declines without the C-Merge layer. This proves that C-Merge has strong capability to effectively capture the correlation between types and items.

*5.4.3 Double threshold mechanism.* We evaluate the effect of the double threshold mechanism by conducting tests in two settings: full version (Full) and None-Threshold. As shown in Figure 9, the model's performance slightly declines without the double threshold mechanism, indicating that the noise in the word-level and attribute-level weights are limited and can be filtered out by our mechanism.

*5.4.4 M-Merge.* We evaluate the effect of M-Merge in motivation-aware preference prediction by conducting tests in three settings: full version of MASP (Full), only merging items' attributes (V-Merge), and only merging users' attributes (U-Merge). As shown in Figure 10, the full version of MASP outperforms other variants across all K values for all datasets. This is because M-Merge combines current and historical attributes for both users and items, enhancing the representativeness of the user and item embeddings. In addition, U-Merge consistently outperforms V-Merge, which indicates that M-Merge has a greater impact on user representations.

## 6 CONCLUSION

This paper proposes a new SPHP problem and a novel framework MASP for SPHP. We first propose a novel HeterBERT for heterogeneous item presentation. Then we propose a motivation-aware model for user preference prediction. Finally, we propose an MCSG algorithm together with two optimisation strategies for efficient session generation. Extensive tests over three real datasets prove that MASP outperforms the SOTA methods in terms of efficacy.

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

# A APPENDIX

## A.1 Proof of Theorem 1

Here, we present the proof of Theorem 1, which has appeared in Sec. 4.1.1.

PROOF. Suppose type representation $\bar{E}_C \in R^{m \times n}$, the difference among all type representations can be computed by:

$$D_t(\bar{E}_C) = \sum_{i=1}^{m} \sum_{j=1}^{m} \sum_{k=1}^{n} (e_{ik} - e_{jk})^2. \tag{7}$$

The variance of $\bar{E}_C$ is computed by:

$$Var(\bar{E}_C) = \frac{1}{n} \sum_{j=1}^{m} \sum_{k=1}^{n} (e_{jk} - \mu)^2 = \frac{1}{n} \sum_{j=1}^{m} \sum_{k=1}^{n} (e'_{jk})^2, \tag{8}$$

where $\mu = \frac{1}{n} \sum_{j=1}^{m} \sum_{k=1}^{n} e_{jk}$. Thus, we have:

$$D_t(\bar{E}_C) = \sum_{i=1}^{m} \sum_{j=1}^{m} \sum_{k=1}^{n} ((e_{ik} - \mu) - (e_{jk} - \mu))^2$$
$$= 2mnVar(\bar{E}_C) - 2Sum(\bar{E}_C\bar{E}_C^T) + 4mn\mu^2 - 2m^2n\mu. \tag{9}$$

For a mean value $\mu$, a larger $Var(\bar{E}_C)$ indicates the elements are more spread out from the mean. Since type vectors are normalized, the equation $Sum(\bar{E}_C\bar{E}_C^T)$ can be regarded as a measurement for vector similarity. When $Var(\bar{E}_C)$ increases, $Sum(\bar{E}_C\bar{E}_C^T)$ declines. Thus, the value of $D_t$ increases with the increase of variance. We conclude that Theorem 1 holds. □

## A.2 ACA algorithm

We propose the ACA algorithm in Sec. 4.2.1 to select candidate category arrangement, as Figure 11 shows.

---

**Input:** $A_q$: activities.
**Output:** $\bar{S}$: candidate arrangement set.
1. $C_q \leftarrow \Phi(A_q)$;
2. Generate $S = \{< c_1, \cdots, c_q >, \cdots, < c_q, \cdots, c_1 >\}$;
3. **if** $\exists s_i \in S$ in database: Assign $P$ based on statistics;
4. **else**: **for** $\forall A_i \subset A_q$:
5.         $P_i \leftarrow ACA(A_i)$;
6.     $P \leftarrow$ Merge $\{P_i\}$;
7. $\bar{S} \xleftarrow{top-k_1} \{S, P\}$;
8. Return $\bar{S}$.

---

**Figure 11: ACA algorithm.**

Given an activity set, we first map activities to the corresponding categories (line 1) and generate all the possible arrangements $S$, where $|S| = q!$ (line 2). If an arrangement can be found in the database, we can compute the probability score for each arrangement $s_i$ by counting their occurrences (line 3). Otherwise, we recursively apply the ACA algorithm to subsets of $S$ until the generated arrangement candidates can be found in the database (line 5). We conduct merge operations over all arrangement candidates generated from the subsets of $S$, $\{P_i\}$, to form the arrangement candidates (line 6). Finally, the top-$k_1$ arrangements with the highest probability scores as the final candidate arrangements (lines 7-8). Given $q$ activities, the time cost of ACA is $O(q!)$.

## A.3 Proof of Theorem 2

Here, we present the proof of Theorem 2, which has appeared in Sec. 4.3.2.

PROOF. Let $e_{u_t}$ be user feature and $M_t$ contain one embedding $V_0$, we have two items with $E_{v_t} = [V_1, V_2]$ and $\bar{E}_{v_t} = [V_1, \bar{V}_2]$, where $e_{u_t}$, $V_1$ and $V_2$ are similar to $V_0$, while $\bar{V}_2$ is not. By setting $\omega_v$ to 0, we formulate the item feature $e_{v_t}$ by:

$$e_{v_t} = (V_1 \cdot V_0^T \cdot V_1)/\alpha + (V_2 \cdot V_0^T \cdot V_2)/\alpha, \tag{10}$$

where $\alpha = V_1 \cdot V_0^T + V_2 \cdot V_0^T$. Similarly, we can formulate $\bar{e}_{v_t}$ for the other item:

$$\bar{e}_{v_t} = (V_1 \cdot V_0^T \cdot V_1)/\bar{\alpha} + (\bar{V}_2 \cdot V_0^T \cdot \bar{V}_2)/\bar{\alpha}, \tag{11}$$

where $\bar{\alpha} = V_1 \cdot V_0^T + \bar{V}_2 \cdot V_0^T$. We compare their relevance scores $\bar{s} = e_{v_t}e_{u_t}^T - \bar{e}_{v_t}e_{u_t}^T$ by:

$$\bar{s} = \left(\frac{V_1 V_0^T}{\alpha}V_1 + \frac{V_2 V_0^T}{\alpha}V_2\right)e_{u_t}^T - \left(\frac{V_1 V_0^T}{\bar{\alpha}}V_1 + \frac{\bar{V}_2 V_0^T}{\bar{\alpha}}\bar{V}_2\right)e_{u_t}^T$$
$$= \frac{1}{\bar{\alpha}\alpha}[(V_1 V_0^T V_1 e_{u_t}^T - \bar{V}_2 V_0^T \bar{V}_2 e_{u_t}^T)\bar{V}_2 V_0^T + (V_2 e_{u_t}^T - V_1 e_{u_t}^T)$$
$$V_1 V_0^T V_2 V_0^T + (V_2 V_0^T V_2 e_{u_t}^T - V_1 V_0^T \bar{V}_2 e_{u_t}^T)\bar{V}_2 V_0^T]. \tag{12}$$

By Eq. 12, when $V_2$ is more similar to $V_0$ and $e_{u_t}$, we get:

$$V_2 e_{u_t}^T > V_1 e_{u_t}^T, \tag{13}$$

$$V_2 V_0^T V_2 e_{u_t}^T > V_1 V_0^T \bar{V}_2 e_{u_t}^T. \tag{14}$$

Thus, we have:

$$e_{v_t}e_{u_t}^T - \bar{e}_{v_t}e_{u_t}^T > 0. \tag{15}$$

When the most related attribute ($V_2$) of an item is replaced by a dissimilar attribute ($V_1$), the relevance score decreases. Given an item $v_1$ with attribute embeddings similar to $M_t$ and $e_{u_t}$ and an item $v_2$ which does not have the similarity, we iteratively apply this process until $v_1$ has no similar attribute and replaced by attributes from $v_2$. The relevance score of $v_1$ continues to decline. Thus, we can claim Theorem 2. □

## A.4 ILG Algorithm

With Theorem 2, we propose a candidate items generation (ILG) algorithm in Figure 12. Specifically, given a type $c_i$ and a user $u$, we first detect attributes that user likes for type $c_i$ by:

$$A_{u,c_i} = A_u \cap A_{c_i}, \tag{16}$$

where $A_u$ is the attribute set from the user interaction history and $A_{c_i}$ contains attributes belonging to type $c_i$ (line 1). In addition, we sample $N_M$ attributes $A_M$ from $A_{c_i}$, based on their frequency of occurrence. We regard $A_M$ as the supplement to user interests since the user could have no historical interaction with the type $c_i$ (line 2). We construct the final attribute set $\bar{A}_{u,c_i} = A_{u,c_i} \cup A_M$ (line 3). Then, we collect items that contain attributes $\{A_{v_i}\}$, where $A_{v_i} \cap \bar{A}_{u,c_i} \neq \emptyset$, and select $N_{filter}$ items as the candidate set for user-item matching (lines 4-6). Assume there are $m'$ sampled items per type, $q$ types, and $k_1$ arrangements. The time cost of ILG is $O((N_M + |A_{c_i}|) * q) + O(m' * q * k_1)$. Since $N_M$ and $|A_{c_i}|$ are much smaller than $m'$, their impact on the overall complexity is neglectable. Thus, the

primary time cost is $O(m' * q * k_1)$, where $m' \ll m$. With ILG, we can significantly reduce the computation cost. In addition, this process can be performed offline, leading to less time cost in online processing.

---

**Input:** $A_u$: attribute set from the user interaction history.
$\qquad A_{c_i}$: set of attributes belonging to type $c_i$.
**Output:** $S_I$: candidate items.
1. $A_{u,c_i} \leftarrow A_u \cap A_{c_i}$;
2. $A_M \leftarrow$ Sample from $A_{c_i}$;
3. $\overline{A}_{u,c_i} \leftarrow A_{u,c_i} \cap A_M$;
4. **for** $\forall v_i \in V_{c_i}$:
5. $\quad$ **if** $A_{v_i} \cap \overline{A_{u,c_i}} \neq \emptyset$: $S_I \leftarrow v_i$;
6. Return $S_I$.

---

**Figure 12: ILG algorithm.**

## A.5 Greedy-MCSG

Given a split line at the $r$-th row, paths generated in the top part have higher approximate scores since each column of $R^{(i)}$ is sorted in descending order. We remove the items with smaller relevance scores from $R^{(i)}$, reshaping the matrix from $R^{k_2*q}$ to $R^{N_r*q}$, where $N_r$ is the number of remaining items in $R^{(i)}$. Then, we apply MCSG on the remaining item set $\overline{\mathcal{R}}$ to fast generate candidate sessions. We analyze the Yelp dataset by computing the ratio ($Rat_{fa}$) of falsely removed candidates to the final sessions to be returned. As reported in Table 5, at $N_r = 1.25K$ where $K$ is the number of returned sessions in the final result, the $Rat_{fa}$ value is close to 0. Thus, we set $N_r$ to $1.25K$ for a good trade-off between the information loss and efficiency of MCSG.

**Table 5: Information loss of approximation.**

| $N_r$ ($\times K$) | 1.05 | 1.1 | 1.15 | 1.2 | 1.25 | 1.3 | 1.35 |
|---|---|---|---|---|---|---|---|
| $Rat_{fa}$ | 0.35 | 0.24 | 0.18 | 0.12 | 0.05 | 0.03 | 0.01 |

## A.6 Detailed Experiment Setup

**Datasets Details.** Details of these datasets are shown in Table 6.

**Table 6: Statistics of datasets.**

| Statistics | # Users | # Items | # Interaction |
|---|---|---|---|
| Yelp | 32,124 | 23,013 | 24,103 |
| Yelp-L | 162,721 | 150,345 | 219,545 |
| Douban | 12,040 | 34,883 | 86,350 |

**Evaluation Metrics.** The effectiveness of models is evaluated in terms of Hit Ratio (HR@K) and Spearman's rank correlation. HR@K is computed by: HR@K = $\frac{\#HIT@K}{\#\mathbb{R}}$, where $\#HIT@K$ is the number of relevant sessions in top-$K$ returned ones and $\#\mathbb{R}$ is the number of all relevant sessions. In SPHP, we use Spearman's rank correlation to measure the "distance" between the generated category arrangement and the ground-truth category arrangement, which is computed as: $\rho = 1 - \frac{6\sum d_i^2}{n(n^2-1)}$, where $n$ is the length of arrangement and $d_i$ is the difference between the two ranks of each category. We evaluate the efficiency of MASP by the response time of the session generation.

**Implementation Details.** All tests are conducted on an Intel i5, 2.30GHz processor machine with 16 GB RAM and 4 GB NVIDIA GTX 1050Ti graphics card. The source code and datasets are available [6].

## A.7 Effect of Dataset Size

We consider the effect of dataset size on the session planning efficiency. Specifically, we divide each dataset into 5 folds and conduct session planning on them separately. As shown in Figure 13, MASP-OPT is much faster than other methods at different updated data sizes. In addition, as the updated data size increases, the time cost of MASP-OPT increases smoothly while those of the other three methods increase proportionally with the updated data size. Clearly, the time cost increase speed of MASP-OPT is much lower than those of the other models with the dataset size increase. Thus MASP-OPT is most scalable in terms of updated data size.

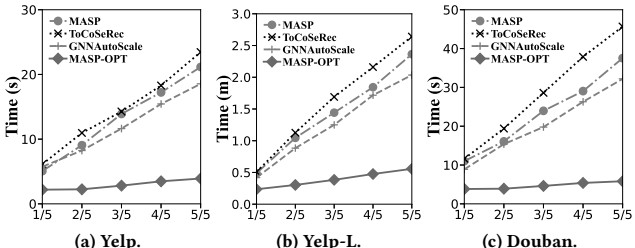

(a) Yelp. $\qquad$ (b) Yelp-L. $\qquad$ (c) Douban.

**Figure 13: Effect of updated data size.**

## A.8 Parameter Tuning

*A.8.1 Effect of $w_T$.* We test the optimal trade-off parameter $w_T$ in Eq. 2 by conducting session planning over three datasets with varying $w_T$ from 0 to 1. As shown in Figure 14 (a), as $w_T$ increases, the HR@20 result of our MASP increases first and declines after an optimal $w_T$ value for all datasets. This proves both word-level and attribute-level correlations affect the effectiveness of MASP. An optimal trade-off is achieved at $w_T$=0.6. Thus, we set the default $w_T$ to 0.6.

*A.8.2 Effect of Word-level Noise Filtering Threshold $th_1$ in HeterBert.* We evaluate the effect of word-level noise filtering threshold $th_1$ in HeterBert to the effectiveness of MASP. We test MASP at different $th_1$ values over three datasets in terms of HR@20. In the test, the $th_1$ value varies from 0 to 1. Figure 14 (b) reports the HR@20 values at different $th_1$. Clearly, the MASP model's performance increases first, achieving the best HR@20 when $th_1 = 0.1$ for all datasets. This is because more word-level noises are filtered out, thus the word-level correction among cleaned words can be better captured in the HeterBert model, leading to higher effectiveness of MASP. With the further increase of $th_1$, the HR@20 result declines. This is because a large $th_1$ threshold incorrectly filters out some word-level word correlation, which causes the information loss. Thus, we set the default $th_1$ to 0.1.

---

[6]https://anonymous.4open.science/r/CODING-87D2/

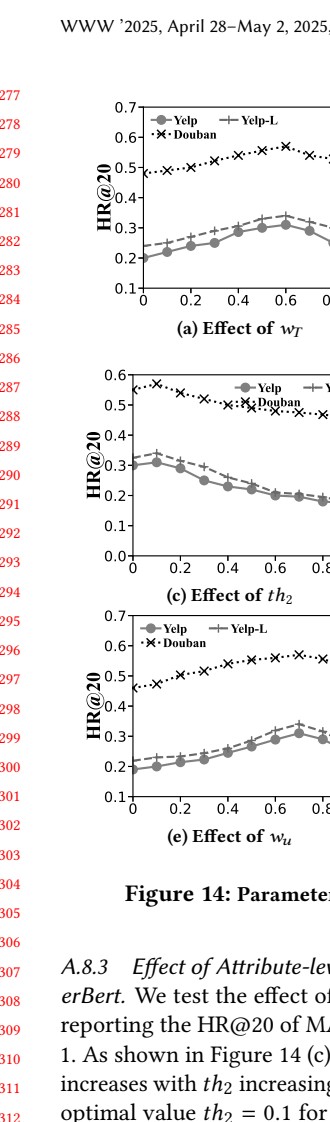

**Figure 14: Parameters vs. the effectiveness of MASP.**

*A.8.3 Effect of Attribute-level Noise Filtering Threshold $th_2$ in HeterBert.* We test the effect of $th_2$ on the effectiveness of MASP by reporting the HR@20 of MASP at different $th_2$ ranging from 0 to 1. As shown in Figure 14 (c), the effectiveness of the MASP model increases with $th_2$ increasing from 0 to 0.1, and then drops after the optimal value $th_2 = 0.1$ for all datasets. This is because applying the double filtering mechanism well filters out the attribute-level noises, which enables the HeterBert model to well capture the word correlations between different cleaned attributes. Meanwhile, extremely filtering after the optimal $th_2$ could filter out some normal attributes, which removes some attribute-level word correlations. Thus, we set the default $th_2$ to 0.1 to balance the quality of attributes and the information loss in HeterBert.

*A.8.4 Effect of $w_v$.* We evaluate the effect of the trade-off parameter $w_v$ in Eq. 4 on the effectiveness of MASP by varying it from 0 to 1 across all datasets. Figure 14 (d) shows that the HR@20 result increases with $w_v$ and reaches the optimal values at $w_v$ =0.8 for all datasets. The HR@20 result drops with the further increase of $w_v$ after $w_v$ =0.8. This proves that both current attributes and historical attributes of items affect the effectiveness of MASP. We set the default $w_v$ to 0.8.

*A.8.5 Effect of $w_u$.* We evaluate the effect of the trade-off parameter $w_u$ in Eq. 5 by varying $w_u$ from 0 to 1. The test results are reported in Figure 14 (e). Clearly, the HR@20 result rises with the $w_u$ increasing, reaches the best values at 0.7 and drops after the best values. This proves that both current attributes and historical

attributes of users affect the effectiveness of MASP. Thus, we set the default $w_u$ to 0.7.

*A.8.6 Effect of Sample Proportion $p_{rop}$.* We evaluate the effect of the sample proportion $p_{rop}$ by searching it from 0 to 1. As shown in Figure 14 (f), as the increase of $p_{rop}$, the HR@20 values increase, reach their peaks at 0.6 for Yelp and Yelp-L, and 0.5 for Douban, and drop with the further increase of $p_{rop}$. This indicates that selectively maintaining the current motivations can better represent items, improving the model's performance. Thus, we set the default $p_{rop}$ to 0.6 for Yelp and Yelp-L, and 0.5 for Douban.

## A.9 Case Study

We conduct a case study to demonstrate the advantages of MASP. We randomly select one user from the Yelp dataset. The user visited a coffee shop (Starbucks), a spa (I touch Day Spa), and a hotel (La Quinta Hotel) in order as the ground truth. Correspondingly, our MASP generates a session plan <Starbucks, I touch Day Spa, Days Inn & Suites> which is represented in blue. MASP accurately predicts two items and provides one hotel that is quite similar in properties, such as location and star rating (in blue arrow). In Table 7, we further report the top-1 arrangement by MASP and the SOTAs on the Yelp data. The user demands a plan for activities (hotel, spa, coffee roasters). First, our ACA model accurately arranges the order of activities. Then, MASP predicts two items correctly while the predictions of GNNAutoScale and TiCoSeRec are completely wrong. This is because GNNAutoScale never considers the inter-item correlation while TiCoSeRec generates a session based on the user's historical records, which is not suitable for handling new trips irrelevant to the user history. Note that "Starbucks" and "I Touch Day Spa" seem to be irrelative but share the same attribute "DogsAllowed", which indicates that MASP can well capture the correlation of heterogeneous items.

**Table 7: Case study: ground-truth and generated plans.**

| Method | Plan |
|---|---|
| Ground-truth | Starbucks, I Touch Day Spa, La Quinta Hotel |
| MASP | **Starbucks**,**I Touch Day Spa**, Days Inn & Suites |
| GNNAutoScale | McDonald's, Elegant Nail, My Place Hotel |
| TiCoSeRec | Dutch Bros Coffee, Tipsy Nails, Holiday Inn Express & Suites |

