# OpenReview forum: "Motivation-Aware Session Planning over Heterogeneous Social Platforms"
_ACM.org/TheWebConf/2025/Conference — WWW 2025 Poster_

### Official Review · Reviewer_ufhV · 2024-12-02

**Novelty:** 5
**Technical Quality:** 5

**Review:**

In this paper, the authors have focused on the problem of session planning on social platforms with different kind of nodes like people and businesses. The paper is well-written and clearly explains the idea. However, I am not convinced about the motivation behind proposing a new problem where Session based recommendation systems (for online platforms) and personalised route planning for offline platforms are already present. Even then, given the prevalence of conversational LLMs like ChatGPT, it would be natural for people to resort to ChatGPT to create a session plan given some activity and neighbourhood/locality information. Hence, a comparison of the proposal with state-of-the-art LLMs were warranted. Practically speaking, the current evaluation results are weak: a hit ratio of 0.3 indicates that only 30% of the suggestions were useful and a user would rather follow their own plan. To properly motivate the problem and demonstrate the effectiveness of the solution, a real user study is required. The authors have presented a case study in Appendix 9, but one case study may not be sufficient for evaluation.

**Questions:**

1. How frequently would people need a service like offline session planning? Won't it be much easier to decide where people would like to go at a particular time period and figuring out a route to minimize the distance travelled.

2. Given the ubiquity of ChatGPT like conversational agent, wouldn't an LLM or fine-tuned version for the task would work better than the proposal? Without a comparison, it is difficult to establish the superiority of the proposal.

3. Aren't the HR@K values low for practical usage?

**Reviewer Confidence:**

3: The reviewer is confident but not certain that the evaluation is correct

**Scope:**

3: The work is somewhat relevant to the Web and to the track, and is of narrow interest to a sub-community

---

### Official Review · Reviewer_sVQ3 · 2024-12-02

**Novelty:** 3
**Technical Quality:** 3

**Review:**

This paper proposes a framework called Motivation-Aware Session Planning (MASP) for session planning across heterogeneous social platforms. The main objective is to optimize the service plan for users on multiple platforms, improving the experience while reducing costs. The experiments are comprehensive, covering a variety of performance and efficiency metrics. Ablation studies and updated data analysis validate the effectiveness and scalability of the model components. Writing is clear and fluency.

a) The theoretical foundation and innovations of the HeterBERT model are relatively weak, and the comparative analysis with traditional BERT is not sufficiently in-depth.
b) While the computational complexity of MCSG is mentioned, there is no in-depth exploration of potential bottlenecks for large-scale real-time applications.
c) Could the authors further discuss the generalization capability of the MASP framework across different types of social platforms (such as social media, e-commerce platforms, etc.)? In the experimental section, the authors tested with three datasets. Are there any biases in these datasets, and could these biases potentially affect the universality of the experimental results?
d) When collecting and using user data, how does the paper address the issue of user privacy protection? Are there any relevant compliance considerations?
e) Users' interests and motivations may change over time. How does the MASP framework handle such long-term changes in user behavior?

**Questions:**

a) Could the authors further discuss the generalization capability of the MASP framework across different types of social platforms (such as social media, e-commerce platforms, etc.)? In the experimental section, the authors tested with three datasets. Are there any biases in these datasets, and could these biases potentially affect the universality of the experimental results?

b) When collecting and using user data, how does the paper address the issue of user privacy protection? Are there any relevant compliance considerations?

c) Users' interests and motivations may change over time. How does the MASP framework handle such long-term changes in user behavior?

**Reviewer Confidence:**

3: The reviewer is confident but not certain that the evaluation is correct

**Scope:**

4: The work is relevant to the Web and to the track, and is of broad interest to the community

---

### Official Review · Reviewer_kpdk · 2024-12-02

**Novelty:** 5
**Technical Quality:** 5

**Review:**

This paper proposes a method to perform session planning, consisting of sequences items constrained by item type. The authors have baptized it as motivation-aware session planning and essentially formulated it as a new type of recommendation problem, which is reasonable to some extent.

The main contribution is the HeterBERT method. I did not find any major technical issues, but it raised me some concerns.

The most obvious is that the paper is extremely dense, in all sections. Because of this, it is very time-consuming to assimilate. Many crucial details are in a 3-page appendix. Clearly the authors have struggled quite a lot to fit the paper within the page limit, and (to their favor) I do not see how to squeeze it further, which begs the comment that this work should be submitted to a journal instead.

A minor issue is that I find the term motivation-aware somewhat excessive, since no real user motivations are modeled. These are simply inferred from the item category.

Finally, I suspect that the authors may have overlooked quite a few contributions in POI recommendation that consider restrictions similar to what the authors define as "motivation". However, since I had not enough time to check thoroughly, I did not consider this as a negative point when scoring the paper.

**Questions:**

Can you please provide details on why standard baselines cannot be used in the experiments?

**Reviewer Confidence:**

3: The reviewer is confident but not certain that the evaluation is correct

**Scope:**

3: The work is somewhat relevant to the Web and to the track, and is of narrow interest to a sub-community

---

### Official Review · Reviewer_NjFq · 2024-12-02

**Novelty:** 5
**Technical Quality:** 5

**Review:**

**Summary**

The paper introduces HeterBERT, which is designed for the session planning task. It offers a comprehensive discussion on the connections and differences between session planning and two closely related tasks: session recommendation and personalized route planning. The authors then present the design of HeterBERT and demonstrate its effectiveness and efficiency through experiments on three public datasets.

**Strengths**

1. The paper studies an interesting and novel task, session planning. The authors provide a clear discussion on the connections and differences between session planning and related tasks, namely session recommendation and personalized route planning.
2. The paper is generally well-written and easy to follow.
3. The proposed method includes several task-specific improvements over the conventional BERT architecture.
4. The code is made available during the review phase.

**Weaknesses**

1. Presentation Issues:
    1. Section 2: This section could benefit from a more structured format, such as bolded headings or subsections for each related field (e.g., session recommendation and route planning).
    2. Figures 2 and 4: The figures are currently low-resolution. Using vector graphics would improve clarity.
    3. Figure 8: Connecting HR@K points with a line is misleading, as observing trends across different K values for a single method is unnecessary -- HR@K always increases with K. A better illustration method, such as a table or bar chart, may be more appropriate.
2. Reproducibility:
    * While the code is provided, it is difficult to correlate with the methods described in the paper due to inconsistencies, such as differing model names.
    * Including scripts to reproduce the results would significantly enhance usability, providing a clear starting point for readers.

**Questions:**

Please refer to "Weaknesses" in "Review" for more details.

**Reviewer Confidence:**

1: The reviewer's evaluation is an educated guess

**Scope:**

4: The work is relevant to the Web and to the track, and is of broad interest to the community

---

### Official Review · Reviewer_4d2q · 2024-12-04

**Novelty:** 3
**Technical Quality:** 3

**Review:**

By proposing the Motivation-Aware Session Planning (MASP) framework, this paper aims to make the first attempt to study the problem of session planning over heterogeneous platforms. In MASP, a HeterBERT model is developed to handle the heterogeneity of items at both type and attribute levels. Then the dynamic user preference is predicted by using the motivations behind user activities, and the sessions with multiple constraints are returned by designing an efficient multi-constraint session generation algorithm (also with optimization techniques).

The reviewer has the following suggestions.
More SOTA related works can be compared to clarify the contributions and differences between the proposed notions and previous ones. The authors are encouraged to compare the “motivation” used in this paper and the “intention” used in [SIGIR24].
[SIGIR24] Zhu Sun, Hongyang Liu, Xinghua Qu, Kaidong Feng, Yan Wang, and Yew Soon Ong. 2024. Large Language Models for Intent-Driven Session Recommendations. In Proceedings of the 47th International ACM SIGIR Conference on Research and Development in Information Retrieval (SIGIR '24). Association for Computing Machinery, New York, NY, USA, 324–334.

The authors are encouraged to compare the proposed model with other multi-modal ones such as the multi-modal session-based recommendation (MMSBR) proposed in [TKDE24].
[TKDE24] X. Zhang, B. Xu, F. Ma, C. Li, L. Yang and H. Lin, "Beyond Co-Occurrence: Multi-Modal Session-Based Recommendation," in IEEE Transactions on Knowledge and Data Engineering, vol. 36, no. 4, pp. 1450-1462, April 2024.

The authors are encouraged to consider multiple modalities in the proposed model architecture as in [TKDE24]. For instance, the motivations could be exploited with prompts for LLMs (as the way in [SIGIR24]). Some motivations can also be presented with images.

More experiments can be conducted to reveal the superiority and necessity of MASP. The authors are encouraged to include more baselines that exploit motivations or intentions in different ways (e.g., [SIGIR24]). The authors are encouraged to conduct case studies to show how motivations are necessary for session planning.

Scallbilty. The use of recurrence positional encoding and type attention layers increases computational overhead, especially when handling large-scale datasets with diverse item attributes and types. The proposed Multi-Constraints Session Generation (MCSG) is inherently complex, as it involves building and evaluating q-partite graphs for candidate arrangements. This makes scalability a challenge for real-time applications.

Generalization. While MASP constructs user profiles based on historical and neighbor attributes, it may oversimplify dynamic user preferences. Real-world user behaviors can be more complex and influenced by contextual factors not captured in the model.
The method prioritizes attributes and types based on occurrence frequency and predefined constraints, which could introduce biases in session generation, neglecting less frequent but potentially important attributes.

The ILG (Item List Generation) and Greedy-MCSG optimizations improve efficiency but sacrifice effectiveness by potentially filtering out relevant items and reducing correlation evaluations between heterogeneous items.

**Questions:**

Please see the detailed comments above.

**Reviewer Confidence:**

4: The reviewer is certain that the evaluation is correct and very familiar with the relevant literature

**Scope:**

3: The work is somewhat relevant to the Web and to the track, and is of narrow interest to a sub-community